# Design, Analysis, and Modeling of an Isolated Constant-Current to Constant-Voltage Converter in Cabled Underwater Information Networks

**Zheng Zhang**, **Xuejun Zhou, Xichen Wang * and Tianshu Wu**

College of Electronic Engineering, Naval University of Engineering, Wuhan 430033, China

* Correspondence: hjgcwxc@163.com; Tel.: +86-180-6254-8837

**Abstract:** The underwater electric energy conversion modules are the equipment for distributing electric energy to the electric instruments in the cabled underwater information networks (CUINs), which are also related to the normal and stable operation of the underwater systems, the energy foundation and an important part of the network platform. Based on the classic push-pull high frequency switching circuit of constant voltage conversion, an isolated constant-current to constant-voltage (CC–CV) converter based on pulse width modulation (PWM) feedback was proposed in this paper. Through the theoretical analysis of the working principle of the CC–CV converter and the modeling of the converter circuit based on the state space method, it was verified that the designed converter can achieve the constant current to constant voltage transformation by high-frequency PWM feedback. Based on Saber simulation software, the circuit model of the CC–CV converter was built, and the feasibility of the design was proved by simulation analysis. According to the isolated high-frequency switching CC–CV converter circuit, the physical prototype was made. By adjusting the resistance value of the output load, the constant voltage (CV) characteristic of the output voltage of the converter was analyzed. Furthermore, the results show that the output voltage of the designed CC–CV converter is stable and the ripple of output voltage is less than 5%, when the input current or output load resistance is adjusted under the CC output status, which has the CV output characteristic. The CC–CV converter has good rationality and feasibility, and is suitable for the future development of the constant current remote power supply system of CUINs.

**Keywords:** cabled underwater information networks (CUINs); constant current; short-circuit grounding high-impedance fault; fault diagnosis; point location; reliability

## 1. Introduction

Over the last several decades, observation of oceans with traditional shipborne and airborne mobile platforms, exploration [1,2] by underwater unmanned vehicles (UUVs), satellite telemetry and other technological approaches has only explored and researched the changes of the epicontinental ocean in a one-sided and intermittent manner, which limited perspective and concepts connected to oceanic observation, understanding and development [3]. Long-term, real-time observation [4] is the requirement of the development of contemporary earth science, meanwhile, varying scientific research needs to reveal the mechanism with the process research. With the development of marine science and technology, in order to explore and understand the ocean and promote the development of marine science, the marine science community has proposed the third platform for observing the ocean [5,6]; that is, cabled underwater information networks (CUINs).

CUINs have eliminated the limitations of time and space in marine research methods, with submarine cables and node equipment connected on the seafloor, which is the information and

power transmission network. It can provide rich power and high-data bandwidth, with standardized connection, transmission and monitoring protocols, as well as unified information, power transmission methods and device interfaces, which are convenient for hooking various observation instruments. CUINs can support multi-element, multidisciplinary, large-scale, long-term, real-time deep-sea science experiments and ocean observations [7–9], which are free from the limitations of energy, weather, ships, data delays and personal safety. In addition, it can meet people's research requirements for long-term, continuous and all-weather observation of the ocean, which has become the focus and inevitable choice for countries to implement marine observation and scientific development [10].

The underwater remote power supply system is the energy foundation of the CUINs, and the performance of the remote power supply system determines whether the CUINs can operate normally. Currently, there are two major types of power supply system in use internationally and both are direct current (DC). In the first type, the trunk cable operates at a nominally constant voltage (CV) [11–13] and all nodes are parallel connected using seawater as a return; an example of that system is the North East Pacific Time-Integrated Undersea Networked Experiments (NEPTUNE- Canada) [14–16]. In the second type, the trunk cable operates in a constant current (CC) mode [17]. An example of that system is the Dense Ocean Network for Earthquakes and Tsunamis (DONET) [18,19]. The system using CC mode has good self-healing properties that protect against short-circuit grounding faults (parts of the system can continue to operate through a fault), and the system does not need high-medium-low voltage conversion, which has the advantages of modularization and standardization of the underwater electric energy conversion modules. In the CC remote power supply system of CUINs, the current in the trunk can be branched by the electric energy branching unit (EEBU) in the underwater primary nodes (PNs), not only keeping the current in the trunk constant, but also the secondary nodes (SNs) on the branch cables can be supplied in a CC or CV power supply mode. The underwater EEBU is mainly composed of one or more isolated high-frequency switching CC converters [9,11], which can be mainly divided into a constant-current to constant-current (CC–CC) converter and a constant-current to constant-voltage (CC–CV) converter. The CC–CC converters are mainly suitable for CUINs with complex topologies, long branches and high power requirements. When the branch cable is short, the energy demand is low, or when the PNs need to take power from the trunk cable to supply power to some transmission and observation instruments, the CC–CV converters are needed for power supply distribution.

The CC–CV converter is one of the core pieces of power supply and distribution equipment of the CC remote power supply system. It is necessary to carry out analysis, design and research to obtain an efficient and reliable converter. Asakawa et al. [20,21] proposed a novel electric energy branching converter for the trunk CC remote power supply system of the ARENA observation network. K. Kawaguchi et al. [22,23] proposed a novel type of power converter for the termination units (TUs) distribution system of the DONET underwater network, and in order to prevent the observation network from the unstable condition of a secondary load; the power distribution system has a balanced converter mechanism that equalizes the TU power consumption constantly while corresponding to the change of secondary load, which has high reliability. Based on the high reliability and low operability requirements, a localization algorithm about the NEPTUNE underwater network for identifying faults in trunk cables by measuring the nonlinear equations of the main line faults at the shore station (SS) was proposed [24]. Lyu F. et al. [25–28] proposed a full-bridge DC converter based on duty-cycle-overlap control (DCOC), which achieved a bipolar positive and negative power supply consisting of eight high-frequency switching devices. Chitta et al. [29] achieved underwater wireless CC–CV transformation by creating an inductive power transfer (IPT) with series compensated coupled coils. Wang et al. [30,31] performed steady-status analysis on the topology of series resonant converters (SRCs) with CC input, and fabricated the hardware of the CC converter, and verified the design. Orekan et al. [32] conducted an analysis in depth of the design of underwater wireless power transfer (UWPT) system, and proposed a method to optimize the coil design to achieve efficient underwater wireless power supply. Saha et al. [33] proposed a CC–CV conversion circuit that uses a parallel

resonant converter (PRC) to achieve a wide range of outputs. Xuewei P. [34] described the traditional circuit systematically. Currently, due to the blockade restrictions of power conversion technology in underwater CC remote power supply systems, the research on underwater electric energy conversion modules of remote power supply system for CUINs is still in its infancy stage, and there are few studies on the CC–CV converters for a CC remote power supply system.

In this paper, theoretical research is carried out on the power demand of underwater electrical energy transformation of CC to CV and the terminal instruments in a CC remote supply system. A novel isolated CC–CV converter based on high-frequency pulse with modulation (PWM) feedback is proposed. A state space method is used to theoretically derive the converter circuit to prove the rationality of the design theory; the Saber simulation software is used to build a CC–CV converter circuit model to prove the feasibility of the design model; a physical prototype is made to verify the reliability of the design circuit. The designed CC–CV converter provides power supply and distribution equipment for the CC remote power supply system of CUINs, which ensures the efficient and reliable operation of CUINs and provides technical support for the trunk construction of CC remote supply system.

## 2. Design Concept of CC–CV Converter Circuit

Currently, there are three main ways to obtain CV piezoelectric energy DC directly from the trunk, which is shown in Table 1. The first type, the power supply of these instruments is based on zener diodes in series connection with the trunk cable. However, with the rise of the current in the trunk, the heat flux of these zener diodes increases rapidly, bringing great threat to the reliability of the system. The second type is in the CV remote supply system, which converts the DC high voltage provided by the SS into the operating voltage of the observation instrument through a high-medium-low two-stage voltage transformation. However, this method has a high withstanding voltage requirement for equipment components; high-voltage long-term operation causes low reliability of the system. The third is to design a CC converter to perform branch power from the trunk, and to deliver CC or CV to the branch. However, the design of the CC converter, especially the CC–CV converter, currently, has poor isolation and anti-interference performance, and most converters cannot achieve the adjustment of the output voltage.

**Table 1.** Comparison of three main ways to obtain constant voltage (CV) piezoelectric energy DC.

|  | Advantages | Disadvantages |
|---|---|---|
| Get power based on zener diode | Simple structure; easy to implement | Only suitable for small current systems; low power |
| Get power based on high-medium-low two-stage voltage transformation | Easy to implement; good research foundation; high voltage resistance; high power | Low reliability; poor standardization; poor self-healing |
| Get power based on CC–CV converters | Standardization; modularity; high reliability; self-healing; high power; easy to expand | Less research foundation; less reference prototype |

The SS mainly includes shore power feeding equipment (SPFE), transmission equipment and monitoring system center. The underwater power supply and distribution system mainly includes four parts: The power supply unit of repeaters, power supply unit of PNs, power supply unit of SNs and underwater power consumption instruments. The main purpose of the CC–CV converter is for the repeaters and the PNs to take power from the trunk cable to provide operating power for the transmission, monitoring and observation instruments or to input CC into the CV for the instrument operation in the SNs. Schematic diagram of CC remote power supply system is depicted in Figure 1 [2].

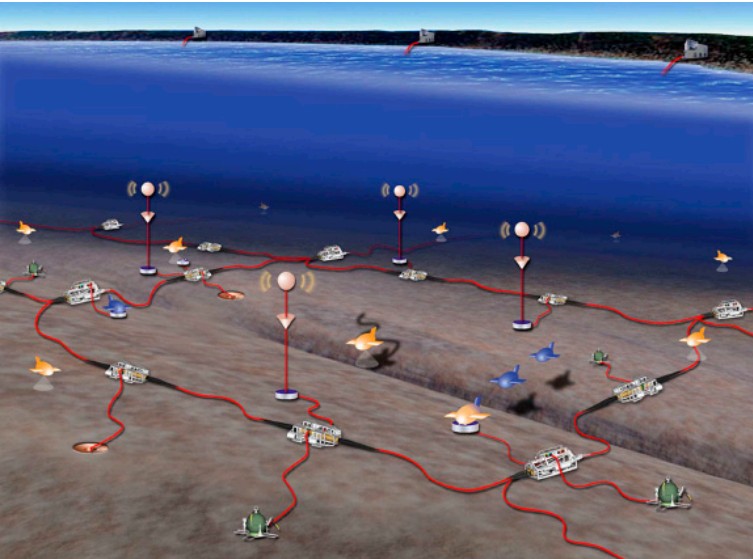

**Figure 1.** Schematic diagram of typical mesh topology CC remote power supply system.

The power supply unit of PNs mainly undertakes the tasks of the branching of CC in the trunk and the power supply to the transmission, control and monitoring instruments in the nodes. The internal structure diagram of the equipment in PNs is illustrated in Figure 2. Current $I_{in}$ flows through the PN, which is branched by the CC–CV converter, and outputs the current $I_{trunk}$ in the trunk, and the CC $I_{branch}$ or CV $U_{branch}$ in the branch. The output current in the trunk is kept constant and equal to the input current; that is, $I_{in} = I_{trunk}$. The constant current to constant voltage transformation is performed by the CC–CV converter in the PNs, and the corresponding standard operating voltages $V_1$, $V_2$ and $V_3$ are provided for the transmission module, the power management control module and the junction box environment monitoring module in the PNs.

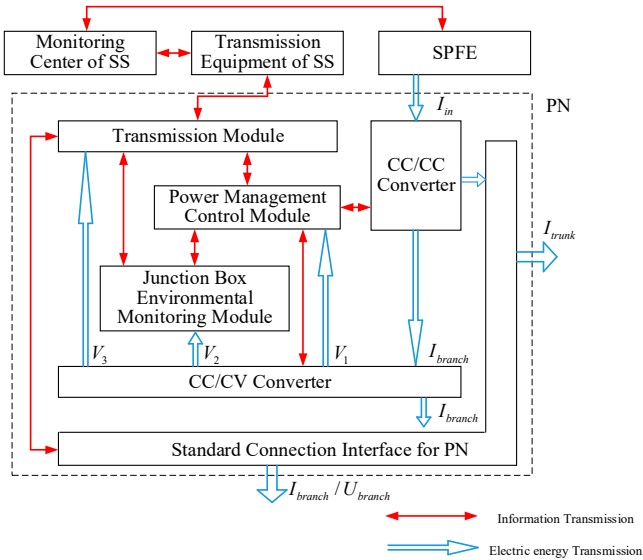

**Figure 2.** Internal structure diagram of the equipment in primary nodes (PNs).

If the PN outputs CC to the SN, the CV transformation is performed by the CC–CV converter in the SN. On the one hand, the rated voltage $V_1 \sim V_3$ is supplied to the internal equipment, and on the other hand, the rated operating voltage $V_4 \sim V_7$ is supplied to the observation instrument through the

standard connection interfaces in SN. Internal structure diagram of the equipment in SNs is shown in Figure 3.

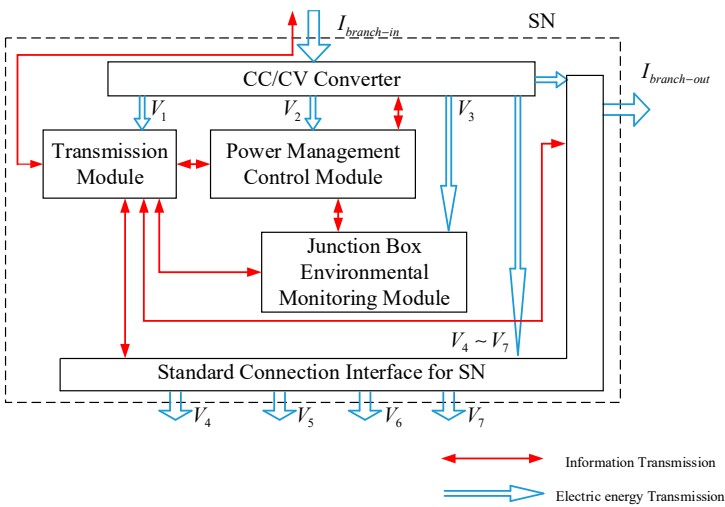

**Figure 3.** Internal structure diagram of the equipment in secondary nodes (SNs).

In summary, as the main part of the EEBU, the CC–CV converter has the main function of transforming the CC input into the CV output, and providing the corresponding operating electric energy for underwater electrical instruments. CC–CV converter should have output stability and good fault self-healing ability. When the connector loses water tightness or equipment failure, it should not affect the upper circuit. Based on the traditional push-pull high-frequency switching circuit of CV converter, an isolated CV converter based on PWM feedback is designed. The circuit schematic diagram is shown in Figure 4, where, $I_{in}$ is the CC provided by SPFE; capacitor $C_1$ supplies power to the output through charge and discharge in the push-pull circuit, and acts as an isolation; $V_{s1}$ and $V_{s2}$ are square wave pulse signals with phase difference of 180°, $S_1$ and $S_2$ are two power switch tubes controlled by pulse signals in turn, and $S_3$ and $S_4$ are power switch tubes controlled by PWM; $R_s$ is a buffer resistor in a push-pull circuit; $Tr$ is a high frequency transformer with equal turns on both sides, $N_1$ and $N_2$ are the primary ends of the transformer, and $N_3$ is the secondary end of the transformer; The output filter circuit is a π-type filter circuit composed of the capacitor $C_2$, an inductor $L_1$ and capacitor $C_3$; $I_{out-trunk}$ is the output current in the trunk of the CC–CV converter mains circuit; $R_{load}$ is the total output total load, and the current $I_{out-load}$ flowing through $R_{load}$ is set as the branch output current; $R_{p1}$ and $R_{p2}$ form a voltage acquisition circuit.

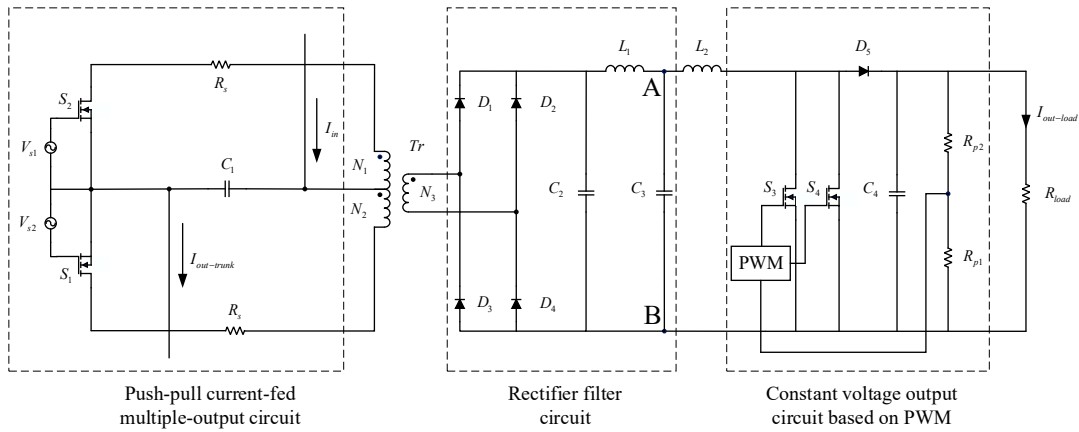

**Figure 4.** Schematic diagram of isolated output-voltage adjustable constant-current to constant-voltage (CC–CV) converter circuit based on pulse width modulation (PWM) feedback.

In order to analyze the steady-status characteristics of the CC–CV converter, it can be assumed that the components in the circuit are ideal components, in which, the voltage drop of the switch tube is zero when the switch is turned on, and the leakage current is zero when the switch is turned off; the inductance and core loss of the transformer are neglected; the passive components in the circuit are linear and time-invariant.

## 3. Characteristics Analysis of CC–CV Converter Circuit

The CC–CV converter circuit can be divided into two parts: (a) The CC isolation part, which mainly includes the push-pull current-fed multiple-output circuit and the rectifier filter circuit; (b) the DC constant voltage conversion part, which mainly includes the constant voltage output circuit based on PWM feedback.

### 3.1. Principle and Characteristic Analysis of CC Isolation Circuit

In the steady-status of the CC–CV converter, since $V_{s1}$ and $V_{s2}$ are square wave pulse signals with a phase difference of 180°, which control switching tubes $S_1$ and $S_2$ with alternated turn-on, the operating process of the circuit is divided into four stages.

Suppose $T_s$ denotes a switch period, and $t_1$ is the start time of one period, at which point $S_1$ starts to conduct, whereas $S_2$ is cut off; at time $t_2$, $S_1$ changes from on to off, and $S_2$ is still off. At time $t_3$, $S_2$ changes from off to on whereas $S_1$ is turn off. At time $t_4$, $S_2$ changes from on to off, and $S_1$ is off. $t_5$ is the end time of this period, and at that moment, both $S_1$ and $S_2$ are turned off. According to the operating status of the CC isolation circuit, ideal waveforms of the voltages of the main components in the circuit in one period can be obtained and depicted in Figure 5.

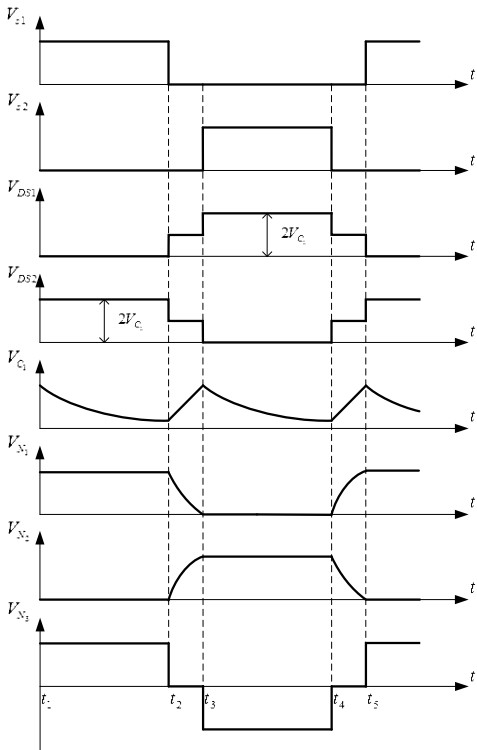

**Figure 5.** Principle waveforms of the voltages of the gate and between the source and the drain of power switch tubes $S_1$ and $S_2$; the primary and secondary ends of the transformer $Tr$ and the capacitor $C_1$ with respect to time variation in the constant-current isolation circuit.

In Figure 5, $V_{s1}$ and $V_{s2}$ are square wave pulse signals of two power switch tubes $S_1$ and $S_2$ with phase difference of 180°; $V_{DS1}$ and $V_{DS2}$ are voltages between the source and the drain of $S_1$ and $S_2$;

$V_{C1}$ is the voltage between two terminals of capacitor $C_1$; $V_{N1}$, $V_{N2}$ and $N_3$ are the voltages of primary and secondary ends of the transformer.

As shown in Figure 5, operation of the circuit can be divided into four phases, which can be summarized into two states: Closed status and open status. The closed status is that the switch tube $S_1$ or $S_2$ is closed; that is, the period $t_3 \sim t_4$ in the operating waveform diagram (the switch tube $S_1$ is turned on, the switch tube $S_2$ is turned off) and the period $t_3 \sim t_4$ (the switch tube $S_2$ is turned on, and the switch tube $S_1$ is turned off);

The open status is when the switch tubes $S_1$ and $S_2$ are both switched off, which is the dead time in the operating period, corresponding to the period $t_2 \sim t_3$ and the period $t_4 \sim t_5$ in the operating waveform diagram.

Because of the mutual inductance of transformer, the reverse voltage of each switch tube at the primary end of transformer $Tr$ in the open-status is half of the primary voltage. The difference is only that it flows through different transformer coils, and the output current direction of the corresponding transformer secondary is opposite.

In the closed status, the operating mechanism is the same in the two periods. Due to the mutual inductance of transformer, the reverse voltage of each switch in the primary end of the transformer $Tr$ is half of the primary voltage. The difference, between $S_1$ and $S_2$, is only flowing through different transformer coils $N_1$ and $N_2$; meanwhile, the output current of the secondary end of corresponding transformer is opposite. When the circuit is in the closed status of period $t_1 \sim t_2$, the capacitor $C_1$ of the primary end of the transformer $Tr$ is discharged, the pulse driving signal $V_{s1}$ is in a high level status, the switching tube $S_1$ is turned on, and the voltage $V_{DS1}$ between the Drain to Source (DS) poles of $S_1$ is zero. The dotted terminal of transformer coils $N_1$ is negative electrode.

When the circuit is in the closed status of period $t_3 \sim t_4$, the pulse driving signal $V_{s2}$ is in a high level status, the switching transistor $S_2$ is turned on, the voltage $V_{DS2}$ between the DS poles of $S_2$ is zero is zero, the dotted terminal of transformer coils $N_2$ is negative electrode. In the rectifier filter part, the rectifier diodes $D_2$, $D_3$ and $D_1$, $D_4$ are in the status of closed and open respectively, the capacitors $C_2$, $C_3$ and the inductor $L_1$ store energy, whereas the equivalent circuit diagram of the constant current isolation circuit is shown in Figure 6.

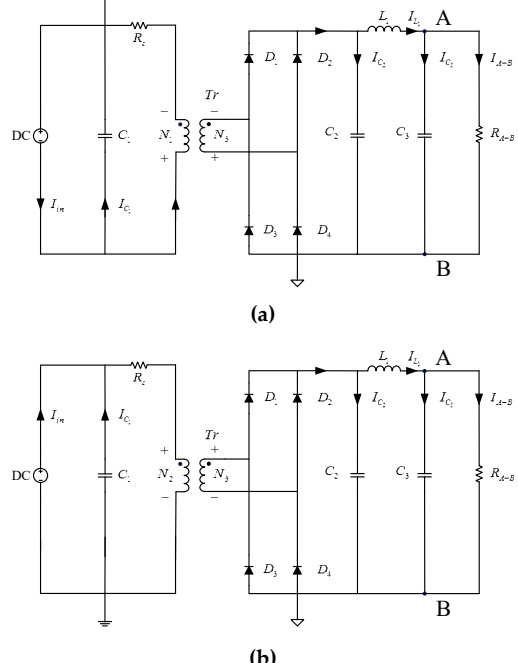

(a)

(b)

**Figure 6.** Equivalent circuit diagram of closed status of constant current isolation circuit: (**a**) Equivalent circuit diagram during $t_1 \sim t_2$ period; (**b**) equivalent circuit diagram during $t_3 \sim t_4$ period.

Where, $i_{C_1}(t)$, $i_{C_2}(t)$, $i_{C_3}(t)$, $i_{L_1}(t)$ and $u_{C_1}(t)$, $u_{C_2}(t)$, $u_{C_3}(t)$, $u_{L_1}(t)$ are the currents and voltages on capacitors $C_1$, $C_2$, $C_3$ and inductance $L_1$, respectively; $i_{A-B}(t)$ is the equivalent output current of isolation circuit, $R_{A-B}$ is the equivalent impedance of DC constant voltage conversion circuit, and $u_{A-B}(t)$ is the equivalent voltage between points A and B.

According to Kirchhoff's current law (KCL) and Kirchhoff's voltage law (KVL), we can obtain:

$$\begin{cases} i_{C_1}(t) = C_1 \frac{d}{dt} u_{C_1} = \frac{u_{C_1}(t) - u_{C_2}(t)/m}{R_s} - i_{in}(t) \\ i_{C_2}(t) = C_2 \frac{d}{dt} u_{C_2}(t) = m\left(\frac{u_{C_1} - u_{C_2}/m}{R_s}\right) - i_{L_1}(t) \\ i_{C_3}(t) = C_3 \frac{d}{dt} u_{C_3}(t) = -\frac{u_{C_3}(t)}{R_{A-B}} + i_{L_1}(t) \\ u_{L_1}(t) = L_1 \frac{d}{dt} i_{L_1}(t) = u_{C_2}(t) - u_{C_3}(t) \end{cases} \tag{1}$$

Assuming that state vector $x(t) = [u_{C_1}(t), u_{C_2}(t), u_{C_3}(t), i_{L_1}(t)]^T$, $u_{C1}(t), u_{C2}(t), u_{C3}(t), i_{L1}(t)$ are state variables, input vector $u(t)$ is $[i_{in}(t)]$, $i_{in}(t)$ is the CC input variable of SPFE. Since the converter has two current outputs, the output current variable in the trunk is $i_{out-trunk}(t)$, and the CC isolation circuit output voltage variable is $u_{A-B}(t)$, then the output vector is $y(t) = [i_{out-trunk}(t), u_{A-B}(t)]^T$; the switching tube duty ratio is D. According to Equation (1), the corresponding status equation and output equations are established for the closed status:

$$\begin{cases} \hat{x}_{close}(t) = \frac{d}{dt} x(t) = A_1 x(t) + B_1 u(t) \\ y_{close}(t) = C_1 x(t) + E_1 u(t) \end{cases} \tag{2}$$

where,

$$\begin{cases} A_1 = \begin{bmatrix} \frac{1}{C_1 R_s} & -\frac{1}{C_1 R_s} & 0 & 0 \\ \frac{1}{C_2 R_s} & -\frac{1}{C_2 R_s} & 0 & \frac{1}{C_2} \\ 0 & 0 & -\frac{1}{C_3 R_{A-B}} & \frac{1}{C_3} \\ 0 & \frac{1}{L} & -\frac{1}{L} & 0 \end{bmatrix} \\ B_1 = \begin{bmatrix} -\frac{1}{C_1} & 0 & 0 & 0 \end{bmatrix}^T \\ C_1 = \begin{bmatrix} 0 & 0 & 0 & 0 \\ 0 & 0 & 1 & 0 \end{bmatrix} \\ E_1 = \begin{bmatrix} 1 & 0 \end{bmatrix}^T \end{cases} \tag{3}$$

Similarly, when the converter is in the open status, both $S_1$ and $S_2$ are turned off, and capacitor $C_1$ is in a charging status. Because the output of SPFE is a constant current source, the voltage across the capacitor rises linearly. Simultaneously, the voltages on the primary and secondary coils of the transformer are all zero, due to the influences of mutual inductance of the transformer, and the voltage $V_{DS1}$ and $V_{DS2}$ between the DS poles of switching transistors $S_1$ and $S_2$ are $V_{C1}$. The output filter circuit releases energy to provide power for the equivalent load $R_{A-B}$. The equivalent circuit diagram of the CC isolation circuit is as shown in Figure 7.

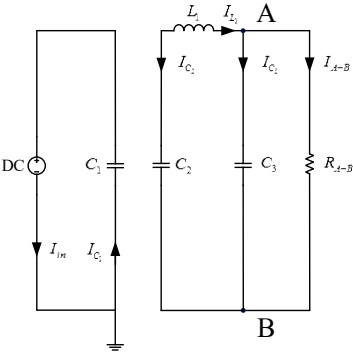

**Figure 7.** Equivalent circuit diagram of open status of constant-current (CC) isolation circuit.

　　　Equally, according to the KCL and KVL, the corresponding state equation and output equation system for the open status can be established,

$$\begin{cases} \hat{x}_{open}(t) = A_2 x(t) + B_2 u(t) \\ y_{open}(t) = C_2 x(t) + E_2 u(t) \end{cases} \tag{4}$$

where,

$$\begin{cases} A_2 = \begin{bmatrix} 0 & 0 & 0 & 0 \\ 0 & 0 & 0 & -\frac{1}{C_2} \\ 0 & 0 & \frac{1}{C_3 R_{A-B}} & \frac{1}{C_3} \\ 0 & \frac{1}{L} & -\frac{1}{L} & 0 \end{bmatrix} \\ B_2 = \begin{bmatrix} -\frac{1}{C_1} & 0 & 0 & 0 \end{bmatrix}^{\mathrm{T}} \\ C_2 = \begin{bmatrix} 0 & 0 & 0 & 0 \\ 0 & 0 & 1 & 0 \end{bmatrix} \\ E_2 = \begin{bmatrix} 0 & 0 \end{bmatrix}^{\mathrm{T}} \end{cases} \tag{5}$$

　　　By the application of direct weighted average method, analyzing the Equations (2) and (4), the average state equation and the average output equation of the CC isolation circuit can be obtained,

$$\begin{cases} \bar{\hat{x}}(t) &= \frac{\mathrm{d}}{\mathrm{d}t}\bar{x}(t) \\ &= D[A_1\bar{x}(t) + B_1\bar{u}(t)] + (1-D)[A_2\bar{x}(t) + B_2\bar{u}(t)] \\ \bar{y}(t) &= D[C_1\bar{x}(t) + E_1\bar{u}(t)] + (1-D)[C_2\bar{x}(t) + E_2\bar{u}(t)] \end{cases} \tag{6}$$

It can be concluded that,

$$\begin{cases} \bar{\hat{x}}(t) = A\bar{x} + B\bar{u} \\ \bar{y} = C\bar{x} + D\bar{u} \end{cases} \tag{7}$$

where,

$$\begin{cases} A = \begin{bmatrix} \frac{D}{C_1 R_s} & -\frac{D}{C_1 R_s} & 0 & 0 \\ \frac{D}{C_2 R_s} & -\frac{D}{C_2 R_s} & 0 & \frac{1-2D}{C_2} \\ 0 & 0 & \frac{1-2D}{C_3 R_{A-B}} & \frac{2D-1}{C_3} \\ 0 & \frac{2D-1}{L} & \frac{1-2D}{L} & 0 \end{bmatrix} \\ B = \begin{bmatrix} \frac{1-2D}{C_1} & 0 & 0 & 0 \end{bmatrix}^{\mathrm{T}} \\ C = \begin{bmatrix} 0 & 0 & 0 & 0 \\ 0 & 0 & 1 & 0 \end{bmatrix} \\ E = \begin{bmatrix} D & 0 \end{bmatrix}^{\mathrm{T}} \end{cases} \tag{8}$$

　　　In the case of DC steady-status, $\bar{\hat{x}}(t) = 0$. By solving the DC steady-status equation, the DC steady-status operating equation of the underwater CC isolation circuit can be obtained as follows,

$$\begin{cases} \bar{x}(t) = -A^{-1}Bu(t) \\ \bar{y}(t) = (E - CA^{-1}B)u(t) \end{cases} \tag{9}$$

　　　Substituting Equation (8) into Equation (9), the relationship between input current and output voltage can be obtained,

$$\begin{cases} \bar{i}_{out-trunk}(t) = Di_{in} \\ \bar{i}_{A-B}(t) = i_{in}(t) \\ \bar{u}_{A-B}(t) = R_{A-B}i_{in}(t) \end{cases} \tag{10}$$

where, $\bar{i}_{out-trunk}(t)$ and $\bar{i}_{A-B}(t)$ are the output currents of the DC steady-status operation of the primary and secondary ends of the transformer; $\bar{u}_{A-B}(t)$ is the equivalent output voltage of the CC isolation circuit.

In summary, the output current of the trunk $\bar{i}_{out-trunk}(t)$ is proportional to the input current $I_{in}$, and the proportional coefficient is the duty cycle $D$. Therefore, in order to maintain the CC characteristics of the constant current remote supply system, it is not feasible to adjust the duty cycle in the CC isolation circuit; meanwhile, the equivalent output voltage of the CC isolation circuit is independent of the duty cycle $D$. When the duty ratio $D$ is changed, the output current $\bar{i}_{A-B}(t)$ is the same as the input current $I_{in}$. The CC isolation circuit can isolate a branch of constant current in the trunk, but according to Equation (10), the output voltage is related to the turn ratio of transformer and input constant current, and it is impossible to achieve constant voltage output by adjusting duty cycle of switch tube. However, the designed constant voltage conversion circuit realizes the constant current to constant voltage transformation with WPM feedback. Therefore, the CC and CV topologies need to be combined.

### 3.2. Principle and Characteristic Analysis of DC Constant Voltage Conversion Circuit

Based on the above analysis, when the CC–CV converter operates steadily, a cycle period of DC constant voltage conversion circuit can be divided into four stages, in which the switching tube $S_1$ or $S_2$ closes, such as the period $t_1 \sim t_2$ (switching tube $S_1$ turns on and switching tube $S_2$ turns off), and the period $t_3 \sim t_4$ (switching tube $S_2$ turns on and switching tube $S_1$ turns off) is in closed status. When the system is in the open status, the switching tube $S_1$ and $S_2$ are open; that is, the dead time in the operating period, corresponding to the period $t_2 \sim t_3$ and period $t_4 \sim t_5$ in the operating waveform diagram, the operating waveform diagram of the DC constant voltage conversion circuit is illustrated in Figure 8.

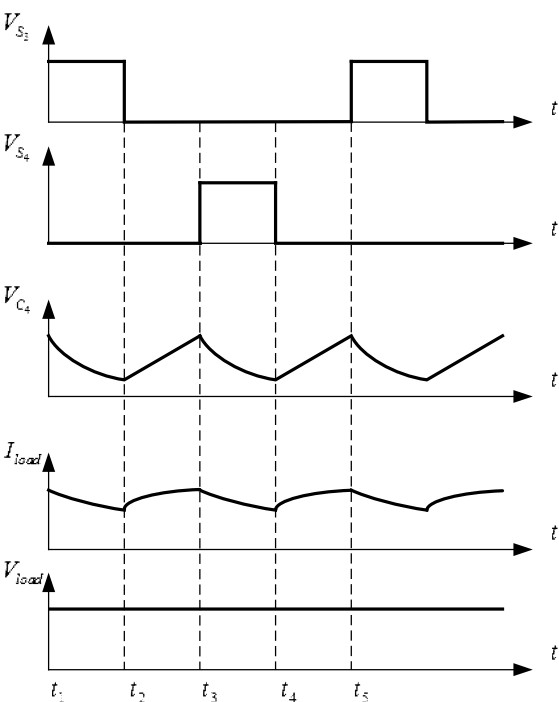

**Figure 8.** Principle waveforms of the voltages of the gate of power switch tubes $S_3$ and $S_4$, the capacitor $C_4$, and the voltage and current of the load $R_{load}$ with respect to time variation in the constant-current isolation circuit.

When the circuit is in closed status, the cc source has no load, and because of the zener diode $D_5$, the capacitor $C_4$ will discharge only to load $R_{load}$. When the circuit is in the open status, the switching

tubes $S_3$ and $S_4$ are open, the CC source supplies power to the load $R_{load}$, and the capacitor $C_4$ is charged. The operating status equivalent circuit of the DC constant voltage conversion circuit is depicted in Figure 9. Because the load is in charged and discharged status, the current direction of capacitor $C_4$ in the closed status is opposite to that in the open status.

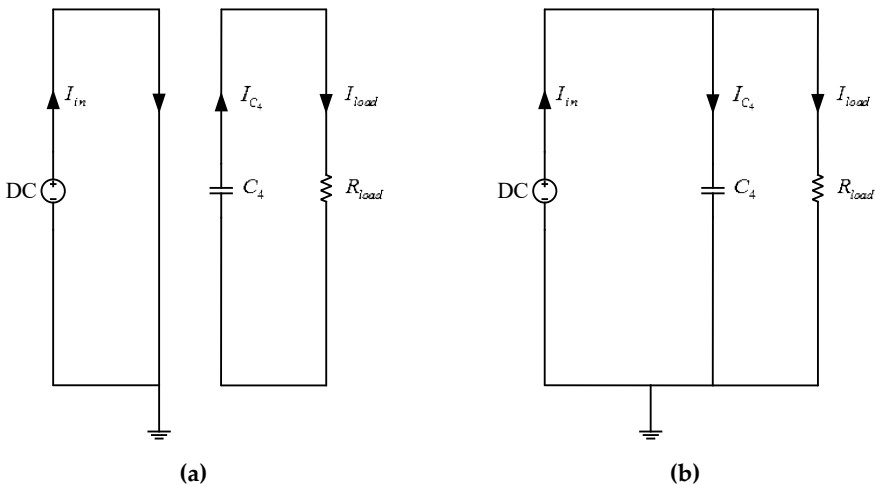

**(a)**             **(b)**

**Figure 9.** Schematic diagram of the equivalent circuit of the working status of the DC constant voltage conversion circuit: (**a**) Equivalent circuit of closed status; (**b**) equivalent circuit of open status.

Where, $i_{C_4}(t)$ and $u_{C_4}(t)$ are current and voltage on capacitor $C_4$ respectively; and $i_{load}(t)$ is the equivalent output current of DC constant voltage conversion circuit.

According to the KCL and KVL, the corresponding state equation and output equation system for the closed status can be established,

$$\begin{cases} C_4\frac{\mathrm{d}}{\mathrm{d}t}u_{C_4}(t) = -\frac{u_{C_4}(t)}{R_{load}} = -i_{load}(t) \\ u_{load}(t) = u_{C_4}(t) \end{cases} \tag{11}$$

Assuming that state vector $x'(t) = [u_{C_4}(t)]$, $u_{C_4}(t)$ is state variable, input vector $u(t)$ is $[i_{in}'(t)]$, $i_{in}'(t)$ is the CC input variable of the DC constant voltage conversion circuit; then, the output vector is $y'(t) = [u_{load}(t)]$; the switching tube duty ratio is $D'$. According to Equation (10), the corresponding status equation and output equations are established for the closed status:

$$\begin{cases} \hat{x}'_{close}(t) = \frac{\mathrm{d}}{\mathrm{d}t}x'(t) = A_1'x'(t) + B_1'u'(t) \\ y'_{close}(t) = C_1'x'(t) + E_1'u'(t) \end{cases} \tag{12}$$

where,

$$\begin{cases} A_1' = \left[-\frac{1}{C_4 R_{load}}\right], & B_1' = [0] \\ C_1' = [1], & E_1' = [0] \end{cases} \tag{13}$$

Similarly, according to the KCL and KVL, the corresponding state equation and output equation system for the open status can be obtained:

$$\begin{cases} \hat{x}'_{open}(t) = A_2'x'(t) + B_2'u'(t) \\ y'_{open}(t) = C_2'x'(t) + E_2'u'(t) \end{cases} \tag{14}$$

where,

$$\begin{cases} A_2' = \left[\frac{1}{C_4 R_{load}}\right], & B_2' = \left[\frac{1}{C_4}\right] \\ C_2' = [1], & E_2' = [0] \end{cases} \tag{15}$$

By the application of direct weighted average method, analyzing the Equations (12) and (14), the average state equation and the average output equation of the DC constant voltage conversion circuit can be obtained,

$$\begin{cases} \bar{\dot{x}}'(t) = \frac{d}{dt}\bar{x}'(t) = D'[A_1\bar{x}'(t) + B_1\bar{u}'(t)] + (1-D')[A_2\bar{x}'(t) + B_2\bar{u}'(t)] \\ \bar{y}'(t) = D'[C_1\bar{x}'(t) + E_1\bar{u}'(t)] + (1-D')[C_2\bar{x}'(t) + E_2\bar{u}'(t)] \end{cases} \tag{16}$$

It can be concluded that,

$$\begin{cases} \bar{\dot{x}}'(t) = A'\bar{x}' + B'\bar{u}' \\ \bar{y}' = C'\bar{x}' + D'\bar{u}' \end{cases} \tag{17}$$

where,

$$\begin{cases} A' = \left[\frac{1-2D'}{R_{load}C_4}\right], \quad B' = \left[\frac{1-D'}{C_4}\right] \\ C' = [1], \quad E' = [0] \end{cases} \tag{18}$$

In the case of DC steady-status, $\bar{\dot{x}}'(t) = d\bar{x}'(t)/dt = 0$. By solving the DC steady-status equation, the DC steady-status operating equation of the underwater DC constant voltage conversion circuit can be obtained—

$$\begin{cases} \bar{x}'(t) = -A'^{-1}B'u'(t) \\ \bar{y}'(t) = (E' - C'A'^{-1}B')u'(t) \end{cases} \tag{19}$$

Substituting Equations (18) into (19), the relationship between input current and output voltage of load can be obtained,

$$\begin{cases} \bar{i}_{load}(t) = \frac{1-D'}{2D'-1}i_{in}(t) \\ \bar{u}_{load}(t) = \frac{1-D'}{2D'-1}R_{load}i_{in}(t) \end{cases} \tag{20}$$

According to Equation (20), when duty cycle $D'$ is more than 50%, the output voltage $\bar{u}_{load}(t)$ of DC constant voltage conversion circuit is not only proportional to resistance of load $R_{load}$ and input current $i_{in}(t)$ of CC source, but also related to duty cycle $D'$ of switch tube in DC constant voltage conversion circuit.

When the resistance of load changes, the input current $i_{in}(t)$ is constant. The voltage $V_{p1}$ can be sampled and adjusted by PWM feedback mode to compare with the reference voltage. The duty cycle of $S_3$ and $S_4$ can be adjusted to maintain the constant output voltage of the system and realize the CC to CV conversion of the system.

$$R_{load} \uparrow \rightarrow V_{load} \uparrow \rightarrow V_{p1} \uparrow \rightarrow PWM \rightarrow D' \uparrow \rightarrow \frac{1-D'}{2D'-1} \downarrow \rightarrow V_{load} \downarrow \tag{21}$$

In summary, the CC isolation circuit achieved electrical isolation of a branch of CC source, which ensures the reliability and anti-interference of the output of system; the CC to CV transformation of the CC source is realized by the DC constant voltage conversion circuit. Combining the CC isolation circuit and DC constant voltage conversion circuit, the result can realize the CV branch output of the current in the trunk of the constant current remote supply system and ensure the current in the trunk is constant.

## 4. Analysis and Research of Modeling Simulation and Verification of Physical Prototype

Based on the theoretical analysis, the simulation model of isolated CC–CV converter proposed in this paper is built with the Saber simulation software, and the feasibility and rationality of the proposed circuit are verified by simulation analysis. Furthermore, the prototype of the physical circuit is fabricated based on the CC remote power supply system; the experimental verification design converter has electrical isolation and CC to CV conversion function.

### 4.1. Modeling and Simulation Analysis of CC–CV Converter Circuit

Based on the references [7–9,34] and the test adjustment of the prototype, assume that some of the electronic component parameters in the circuit are: $C_1 = 3\ uF$, $C_2 = 1\ uF$, $C_3 = 2\ uF$, $C_4 = 100\ uF$, $L_1 = L_2 = 6\ mH$, the buffer resistance $R_s$ between the drain (D) and the source (S) of the switching tubes $S_1$ and $S_2$ is 5 Ω.

As shown in Figure 4, performing current output analysis on the CC isolation circuit, and select three sets of tests for the duty cycle of switching tube $S_1$ and $S_2$ are 60%, 80% and 100% respectively. The current $i_{A-B}(t)$, between nodes A and B, is simulated and analyzed, while the simulation results are as shown in Figure 10.

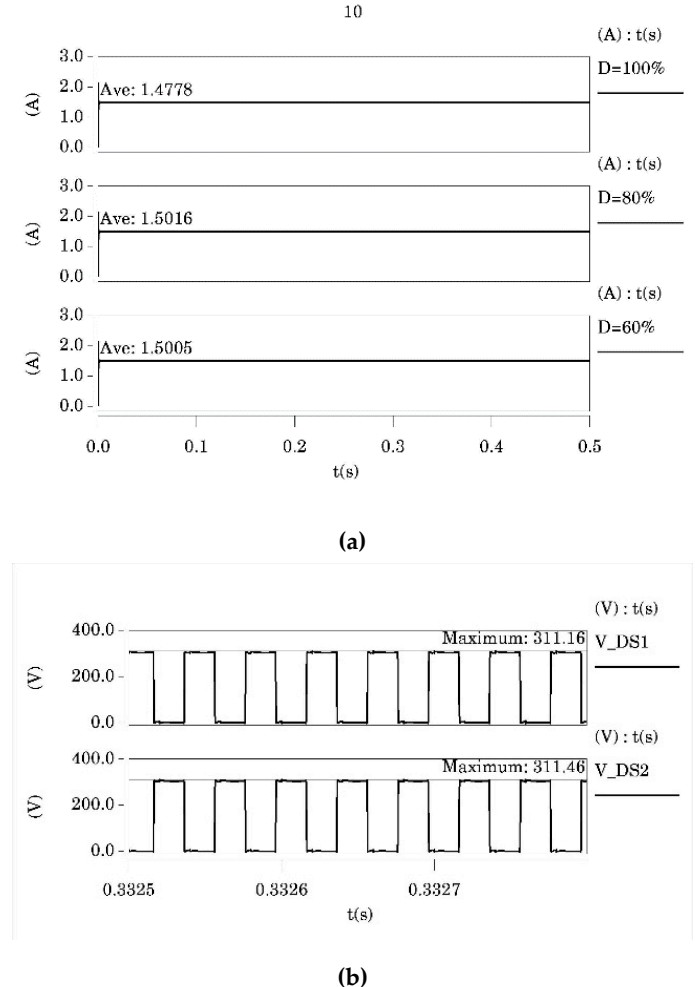

(a)

(b)

**Figure 10.** Schematic diagram of simulation results of constant current isolation circuit: (**a**) Schematic diagram of current simulation between nodes A and B (the duty cycle of switching tubes $S_1$ and $S_2$ are 60%, 80% and 100%); (**b**) schematic diagram of simulation results for voltages of Drain to Source (DS) poles of switching tubes (the duty cycle is 100%).

Figure 10 shows the schematic diagram of the simulation results of the CC isolation circuit under the condition of adjusting duty cycle $D$. Assuming that the input constant current is 1.5 A, the equivalent resistance between loads is adjusted to 100 Ω. When duty cycle changes, the simulation output current of CC isolation circuit is basically the same as that of theoretical analysis. The output current is stable, and the fluctuation of current is less than 1.5% of the theoretical value, which is as shown in Figure 10a. It can be determined that the output current of constant current isolation circuit is DC constant current.

After the converter is in stable operation, the driving circuit controls the switching tubes $S_1$ and $S_2$ to be turned on, and the capacitor $C_1$ is charged and discharged accordingly, and the voltage $V_{C_1}$ is charged and discharged with a small amplitude at a voltage of about 150 V according to the opening and closing of the light-emitting tube. The voltage value of the load $V_{load}$ should be 150 V, and the open-status voltage of switching tubes $S_1$ and $S_2$ should be twice as much as $V_{load}$. As shown in Figure 10b, the voltages of $V_{DS1}$ and $V_{DS2}$ in the open-status are about 300 V, which are substantially the same as the theoretical values. Comparing Figures 4 and 10, obviously, it can be seen that the simulation waveform of the designed constant current isolation circuit is similar to that of the theoretical waveform. The simulation results show that the circuit can achieve constant current isolation output.

When the CC isolation circuit outputs constant current, the DC constant voltage conversion circuit is simulated and analyzed. The voltage $V_{p1}$ at both ends of the sampling resistance $R_{p1}$ is compared with the reference voltage $V_{REF}$, fed back by PWM. The duty cycle of switching tubes $S_3$ and $S_4$ is adjusted to maintain the constant output voltage of the system and realize the CC to CV conversion of the system. Assuming that the load resistance $R_{load}$ varies, set to 100 Ω, 200 Ω, 300 Ω, 400 Ω and 500 Ω, and the sampling resistances $R_{p1}$ and $R_{p2}$ are 2k Ω and 60k Ω respectively, the reference voltage $V_{REF}=$ 5.1 $V$. Having simulation and analysis of the DC constant voltage conversion circuit, the simulation results are as shown in Figure 11.

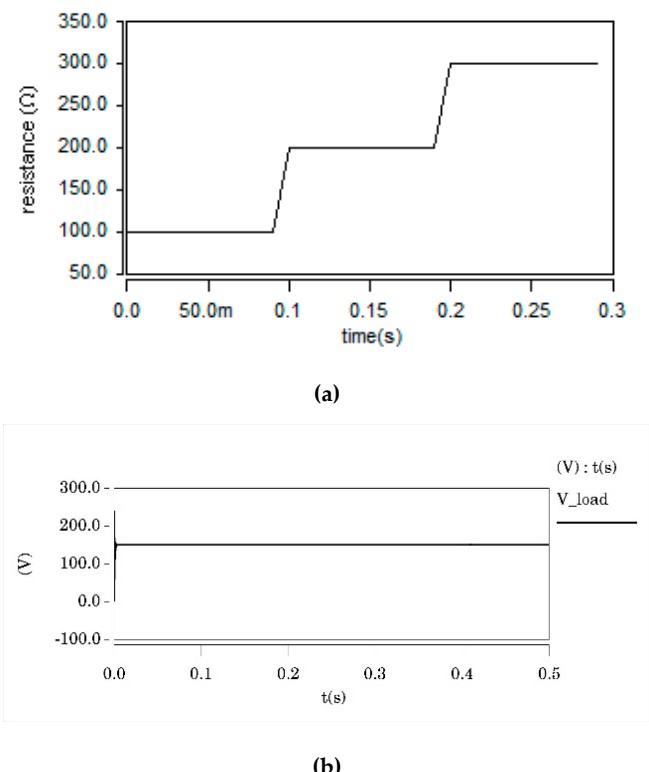

(a)

(b)

**Figure 11.** Schematic diagram of simulation results of the relationship between load voltage and resistance change in DC constant voltage conversion circuit: (**a**) Curve of change in the resistance of load; (**b**) voltage curves of the load ($R_{load}$ = 100 Ω, 200 Ω, 300 Ω, 400 Ω and 500 Ω).

Figure 11 shows the Schematic diagram of simulation results of DC constant voltage conversion circuit in the case of adjusting the resistance of load $R_{load}$. Assuming a constant input current of 1.5 A, as shown in Figure 11a, the output load resistance is $R_{load}$ = 100 Ω, 200 Ω, 300 Ω, 400 Ω, and 500 Ω; the equivalent voltage value at both ends of the load $R_{load}$ is basically the same as the theoretical analysis, and the output current is stable. It can be determined that the output of the DC constant voltage conversion circuit is a DC constant voltage. When the load $R_{load}=$ 100 Ω is set, the driving circuit and the switching tubes $S_3$ and $S_4$ are 180° out of phase, and when the switching tube is

in the closed state, the voltage on the switching tube is zero. When the switch tube is in the open status, $S_3$ and $S_4$ are connected in parallel, and the voltages at both ends of the tubes are the same, which is consistent with the output load voltage, as shown in Figure 12. Comparing the analysis of Figures 5, 11 and 12, the simulation waveform of the DC constant voltage conversion circuit of the designed CC–CV converter is similar to the theoretical waveform; that is, the designed CC–CV converter has CV output characteristics.

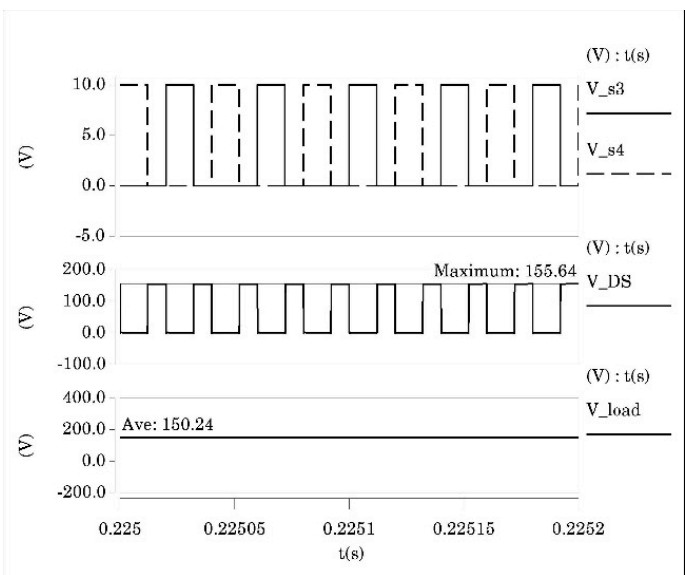

**Figure 12.** Schematic diagram of simulation curves of square wave pulse signals $V_{s3}$ and $V_{s4}$, voltage between the DS poles of $S_3$ and $S_4$ ($R_{load} = 100\ \Omega$) in DC constant voltage conversion circuit.

Through simulation analysis, it can be concluded that the isolated high-frequency switching CC–CV converter can multi-branch the constant input current, which can not only ensure the constant characteristics of the current in the trunk, but also maintain the requirement of future mesh topological structure of the CC network. The output DC constant voltage is stable, which can meet the operating electricity demand of underwater instruments, and provide a new technical method for power supply and distribution of underwater CC remote system in CUINs.

### 4.2. Research on Verification of Prototype of CC–CV Converter

Based on the proposed CC–CV converter circuit design and simulation analysis, a principle prototype converter was built in the laboratory to further verify the rationality and feasibility of the proposed CC–CV converter circuit. The principle prototype of the CC–CV converter is as shown in Figure 13.

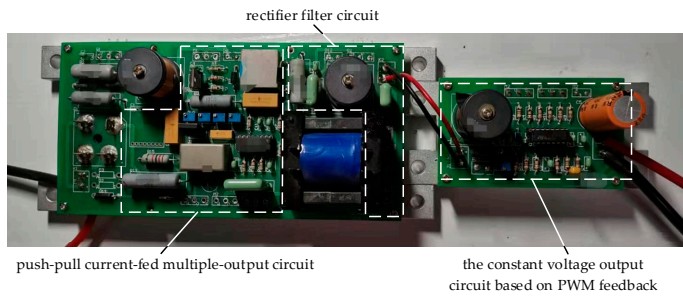

(a)

**Figure 13.** *Cont.*

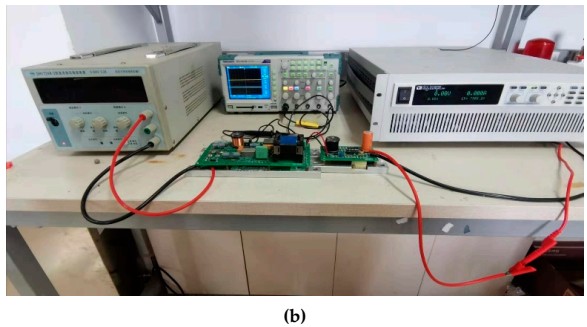

**(b)**

**Figure 13.** Test of CC–CV converter: (**a**) Principle prototype of CC–CV converter; (**b**) construction of laboratory test environment.

According to the designed prototype, the CV output characteristics were verified by experiments.

Adjusting the duty cycle $D$ of the switching tubes in the CC isolation circuit to 80%, the resistance of load value was set to 100 $\Omega$; the resistance values of sampling resistors $R_{p1}$ and $R_{p2}$ were adjusted; the output voltage $U_{load}$ was set to 150 V. Furthermore, the voltage signal curves of the switching tubes in the CC isolation circuit and the DC constant voltage conversion circuit were acquired and analyzed, which are depicted in Figure 14.

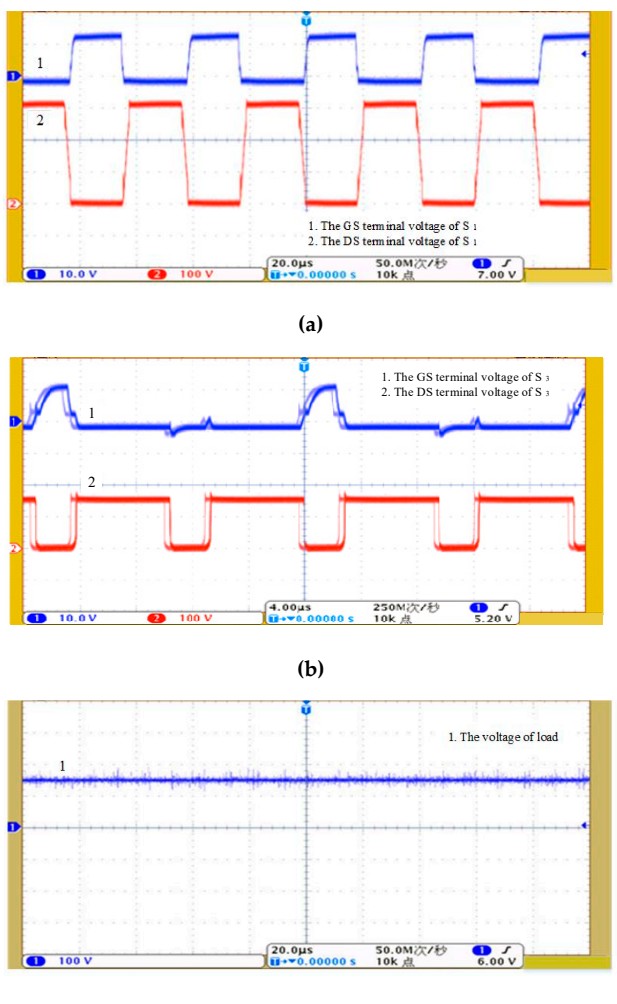

**Figure 14.** Voltage acquisition test curve of Gate to Source (GS) and Drain to Source (DS) poles of $S_1$ and $S_3$, and the output of load $R_{load}$ in CC–CV converter: (**a**) voltage curves between the GS and DS poles of $S_1$; (**b**) voltage curves between the GS and DS poles of $S_3$; (**c**) output voltage curve of load.

Comparing Figure 14a,b with Figures 5 and 8, the voltage acquisition waveform of the prototype is nearly the same as that of the theoretical derivation and simulation analysis. In the CC isolation circuit, when the switching tubes are closed by the square wave pulse signals, the Drain to Source (DS) voltage of the tubes is zero, whereas the DS voltage is twice as much as the voltage of capacitor $C_1$, which is about 300 V, when the tubes are open. By adjusting duty cycle of DC constant voltage converter through sampling circuit and PWM feedback circuit, the output voltage can be constant. As shown in Figure 14c, the output voltage is stable at about 150 V and fluctuates within 5%, which is a good CV characteristic and can meet the requirement for the operating power-stability of underwater electrical instruments.

According to the requirement of the experiment, the stability and practicability of the constant current-constant voltage converter need to be further verified by testing the constant characteristic of the output voltage by changing the resistance value of the load $R_{load}$. Sampling resistance was set, CV output was set at 150 V, output load resistance $R_{load}$ is adjusted, and a constant characteristic test of load polarity output voltage with seven different resistance values was set. The test scenario is depicted in Figure 13b. The experimental results, are shown in Table 2 and Figure 15.

**Table 2.** Experimental test data in the different resistance values of load.

| $R_{load}/\Omega$ | 100 Ω | 150 Ω | 200 Ω | 300 Ω | 500 Ω | 750 Ω | 1k Ω |
|---|---|---|---|---|---|---|---|
| $I_{in}/A$ | 1.500 | 1.504 | 1.503 | 1.499 | 1.503 | 1.501 | 1.500 |
| $I_{A-B}/A$ | 1.482 | 1.491 | 1.496 | 1.498 | 1.498 | 1.503 | 1.509 |
| $I_{out-load}/A$ | 1.499 | 0.998 | 0.748 | 0.509 | 0.303 | 0.199 | 0.149 |
| $U_{in}/V$ | 171.985 | 121.460 | 95.056 | 69.101 | 47.917 | 35.924 | 30.982 |
| $U_{A-B}/V$ | 160.013 | 109.631 | 84.373 | 57.948 | 37.952 | 27.964 | 22.868 |
| $U_{out-load}/V$ | 149.946 | 150.032 | 150.073 | 150.022 | 150.047 | 150.033 | 149.986 |

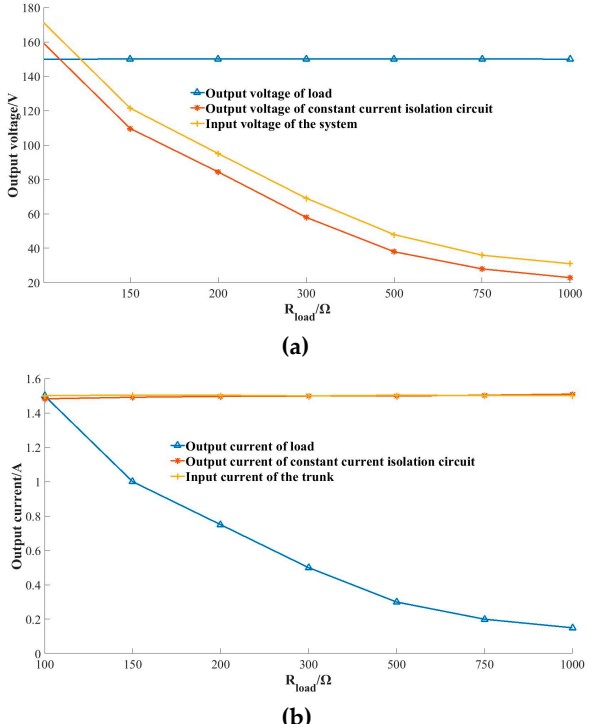

(a)

(b)

**Figure 15.** Curve of relationship between resistance value of load and output voltages and currents of load, CC isolation circuit and the trunk: (**a**) Curve of relationship between load and the voltages of load, CC isolation circuit and the trunk; (**b**) curve of relationship between load and the currents of load, CC isolation circuit and the trunk.

As shown in Table 2, with the increase of load resistance value $R_{load}$, the output voltage of the load $U_{out-load}$ remains unchanged because of the constant voltage output characteristic of the converter; besides, because of the constant current characteristic of the trunk line and the isolation circuit, the input current of the system $I_{in}$ and the output current of the isolation circuit $I_{A-B}$ remain unchanged. According to $P = U^2/R$, the input voltage of the system $U_{in}$, the output voltage of the CC isolation circuit and $U_{A-B}$, the output current of the load decreases with the increase of the load.

As shown in Figure 15a,b, when the value of load resistance changes, the output current of the CC isolation circuit remains unchanged, which is nearly the same as that of the trunk and has good CC characteristics. The output voltage of the DC constant voltage conversion circuit remains unchanged under load changes, and the CV output characteristics of the CC–CV converter are excellent. The input voltage of the CC–CV converter and output current of load are linearly related to the resistance of load. The input voltage value of the SPFE is related to the output power of the load, and the SPFE does not need to output high voltage, while more the underwater equipment is less affected by the high voltage. Furthermore, the system reliability is greatly improved, and the current in the trunk is constant, and the power of cable loss in the whole network is constant, which is beneficial to the system fault determination. As shown in Figure 16, with the decrease of load, according to $P_{load} = U_{load}^2/R_{load}$, the loss of the converter is basically stable. With the increase of load power, the output efficiency of the converter is improved. The designed CC–CV converter has high conversion efficiency, and the loss of the full module conversion is controlled within 30 W. In the condition of a full load, the conversion efficiency exceeds 90%, and the output power of a single converter is up to 230 W.

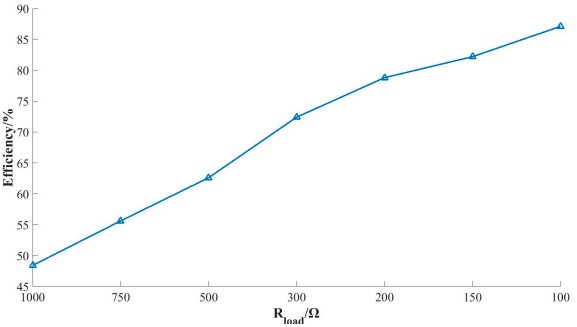

**Figure 16.** Curve of relationship between load and the output efficiency of the CC–CV converter.

The CC isolation circuit and the DC constant voltage conversion circuit are analyzed separately, and Sections 2 and 3 prove that the theory of isolated circuit is feasible. In response to the isolated power converters proposed, the simulation is carried out and the prototype is designed. Through comprehensive theoretical derivation, simulation analysis and experimental verification, it can be verified that the proposed isolated CC–CV converter has reliable and efficient constant current transformation function for a branch, by combining the theory and simulation analysis of a constant current-constant voltage converter and building a test circuit prototype. On the basis of maintaining the CC characteristics of underwater current in the trunk, the CC–CV converter realizes electric energy conversion; the CC–CV converter can provide stable and reliable power supply for branch observation equipment without high-medium-low voltage conversion, and the CC–CV converter has the advantages of standardization, modularization, high power, easy expansion and high reliability. At the same time, the electrical isolation characteristic ensures that the converter is free from electromagnetic interference caused by a branch fault or current fluctuation, and ensures the stability and reliability of the constant current in the trunk. Moreover, through series-parallel redundancy of multiple converters, a high working-power can be stabilized for future high-power underwater observation equipment.

## 5. Conclusions

With the continuous development of cabled underwater information networks, the supply and distribution equipment of remote power supply system has become an important part of system operation and maintenance. Based on Kirchhoff's current law and Kirchhoff's voltage law, this paper carries out the theoretical analysis of the proposed constant current-constant voltage converter. The simulation circuit module is established, and the rationality and feasibility of the converter are verified by analyzing the simulation. The physical prototype of a constant current-constant voltage converter is fabricated, and the test environment is built. Furthermore, the reliability and practicability of the design are verified by changing the magnitude of load resistance and duty cycle. The designed constant current-constant voltage converter, which can be used in the constant current remote power supply system, can provide the required operating electric energy for underwater observation instruments, and has high practical value. Currently, this method has passed the initial testing; in order to test and verify the design and power supply adaptability of the designed converter, the next step would be to build the multi-node prototype test platform for constant current remote supply system, and furthermore, to verify the practical application performance of the converter in this paper through experiments.

**Author Contributions:** Conceptualization, Z.Z. and X.Z.; data curation, T.W.; formal analysis, Z.Z. and X.W.; Writing—Original Draft, Z.Z.; Writing—Reviewing and Editing, X.Z. and X.W.

**Funding:** This work was supported in part by the National Natural Science Foundation of China under Grant number 61871473 and number 61803379, and the Natural Science Foundation of Hubei Province, China under Grant number 4251181107.

**Conflicts of Interest:** The authors declare no conflict of interest.

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
