# Peer review of "Design, Analysis, and Modeling of an Isolated Constant-Current to Constant-Voltage Converter in Cabled Underwater Information Networks"

_electronics, doi:10.3390/electronics8090961_

Round 1
Reviewer 1 Report
The research design is sound and it is presented well. However, the language needs to be improved. There are too long sentences, including the title itself. These should be shortened and written as much smaller sentences. IT makes it hard to follow the content.
Before publication, below points need to be addressed:
1) Novelty: this is not mentioned in the introduction or conclusions. Does this work have some novel design? OR does it contribute to body of knowledge with implementing and analyzing a known CC-CV converter?
2) In introduction, refrain from citing many references at once. e.g. 1-6. Try to spread them around to build your case in a sound way.
3) Also, it is not clear from the paper why CC and CV topologies need to be combined. The paper mentions these topologies as separate designs but then moves on to combine these.
4) Beginning of section 2, it would be helpful to add a table to compare 3 known approaches to obtaining CV and the method presented in the paper. a table with method, pros and cons
5) in section 4, what is simulated and what was the analysis point? These need to be clarified. A topology of the studied system would be beneficial.
6) in 4.1, some values are assumed for components. What are these based on? A reference? practical information? This needs to be added.
7) the uniqueness of this research is about its field of use. in CUINs what is the usual distance between the converter and the power supply and the load? The lab test set up is generic and does not study this impact. Furthermore, for solid state components temperature and pressure are very important. Considering CUINs are located under water in the oceans, what would be the impact on the performance?
Reviewer 2 Report
Please clarify the problem statement in the introduction. It should be highlighted what sets this design/model apart from others and why should this design be adopted. Please avoid word redundancies, such as “distributing and distributing” in lines 11-12, and “sporadic, intermittent” in line 36. Please look out for typing mistakes, such as in line 81, it should be “in” instead of “n”. Please fragment long sentences into smaller ones for easier interpretation, such as lines 84-91. Kindly check for grammatical errors. In line 93, it should be “achieved” instead of “achieve”; in line 94, it should be “creating” instead of “create”; in line 193, it should be “threats/threat” instead of “threaten”; in line 140, it should be “outputs” instead of “output”. Kindly check for punctuation errors. In line 131, a full-stop is expected after “power consumption instruments”. Please make the illustration at Figure 1 much clearer. The choice of colors often influences comprehension. Kindly check all subscripts. In line 163, Vs1 is repeated. The second mention should be Vs2. In line 164, it should be “S3 and S4”. In line 199, it should be “t1 and t2” instead of “t3 and t4”. In line 218, it should be “VDS2” instead of “VDS1”. In line 362, it should be “VDS1 and VDS2” instead of “VDS1 and VDS1”. Please keep note of sentence structures, such as in lines 186-188, either “since” in line 186 or “and” in line 188 should be omitted. Kindly make the figure titles a little more descriptive and self-explanatory, such as the ones in Figures 5, 8, 11, 13, 14. Please maintain the same format in mentioning variables, such as, the derivative of uC1 in the first equation in the equation set (1) should also be indicated as a function of time, just as the other three equations. Kindly edit the equations for = , instead of = (t). This is seen in equation 6 and line 259. Please add a short paragraph after Table 1, to mention the trend of change of the experimental test data, along with the reason of their change. Kindly elaborate what kind of problem of underwater electric energy supply and distribution can this model/design solve, as indicated in lines 474-475. Please improve the quality of all the images. Please incorporate references from standard and well-known sources. Please try to make the conclusion better. Please use standard simulation tools such as MATLAB/Simulink. In line 108, a reference has been cited where the purpose of this paper is stated. This makes the originality of the paper questionable.Author Response
Please see the attachment.

Round 2
Reviewer 2 Report
It appears that the authors have improved the paper based on the comments, except for one: they did not increase the image resolution. Many texts in the figures are not readable. I believe the editorial office will take care before publishing it.
